# Investigation of Stress Corrosion Cracking Resistance of Irradiated 12Cr Ferritic-Martensitic Stainless Steel in Supercritical Water Environment

**DOI:** 10.3390/ma16072585

**Published:** 2023-03-24

**Authors:** Boris Margolin, Natalia Pirogova, Alexander Sorokin, Vasiliy Kokhonov, Alexey Dub, Ivan Safonov

**Affiliations:** 1Central Research Institute of Structural Materials “Prometey” of National Research Center “Kurchatov Institute”, 191015 Saint-Petersburg, Russia; 2Joint-Stock Company “Science and Innovations”, 115035 Moscow, Russia

**Keywords:** supercritical water-cooled reactors, ferritic-martensitic stainless steel, stress corrosion cracking, constant load test, neutron irradiation, oxide layer, microcleavage fracture

## Abstract

The supercritical water-cooled reactors (SWCR) belong to Generation IV of reactors. These reactors have a number of advantages over currently operating WWERs and PWRs. These advantages include higher thermal efficiency, a more simplified unit design, and the possibility of incorporating it into a closed fuel cycle. It is therefore necessary to identify candidate materials for the SWCR and validate the safety and effectiveness of their use. 12Cr ferritic-martensitic (F/M) stainless steel is considered a candidate material for SWCR internals. Radiation embrittlement and corrosion cracking in the primary circuit coolant environment are the main mechanisms of F/M steels degradation during SWCR operation. Here, the stress corrosion cracking (SCC) in supercritical water at 390 and 550 °C of 12Cr F/M steel irradiated by neutrons to 12 dpa is investigated. Autoclave tests of specially designed disk specimens in supercritical water were performed. The tests were carried out under different constant load (CL), temperature 450 °C, and pressure in autoclave 25 MPa. The threshold stress, below which the SCC initiation of irradiated 12Cr F/M steel does not occur, was determined.

## 1. Introduction

WWER and PWR reactors currently have a leading position in the nuclear power industry. WWER reactors use water pressurized up to 16 MPa with a temperature of up to 324 °C as the coolant. Increasing the coolant pressure to 25 MPa and the reactor outlet coolant temperature to 550 °C increases the thermal efficiency of the units (up to 45%) and reduces the environmental impact by reducing heat losses in the thermodynamic cycle from 67% (WWER-1000) to 55% (WWER SWC) [1,2,3,4,5]. Another advantage of SWCR is the possibility of different active core designs: with a thermal neutron spectrum for operation in an open fuel cycle with UO_2_ fuel and with a fast neutron spectrum for operation in a closed fuel cycle with MOX fuel [6]. So, it is important to identify candidate materials for the SWCR and validate the safety and effectiveness of their use.

It has been shown [4,7,8,9,10,11,12] that austenitic stainless steels, F/M steels, titanium zirconium alloys, alumina-forming austenitic (AFA) stainless steel, and oxide dispersion-strengthened (ODS) steel can be used as candidate materials for different components of SWCR. All of these materials have a number of advantages and disadvantages.

In this paper, we consider 12Cr F/M steel developed at our institute as a candidate material for SWCR internals.

It has been shown [4,12,13] that F/M steels are highly susceptible to general corrosion, especially compared to austenitic steels. So, in [8], it was shown that after a 1500 h exposure test of T91 F/M steel in 600 °C SCW, the oxide film thickness exceeded 75 μm. The thickness of the oxide film increases with increasing of the test temperature [8]. The oxide film consists of three layers: the internal diffusion layer (a mix of oxides and matrix), (Fe, Cr)_3_O_4_ spinel, and Fe_3_O_4_ magnetite (outer layer). Since cracking and exfoliating of the oxide film are possible during reactor operation, F/M steels are not very suitable for fuel rod cladding with a thickness comparable to that of the oxide film. At the same time, it has been shown that F/M steels have a much higher resistance to SCC compared to austenitic steels [4,12,13]. Therefore, we consider F/M steels as a candidate material for SWCR internals with a thickness much larger than the oxide film.

Constant loading (CL) tests represent the actual operating conditions of the internal materials for the SCC studies in the best way, but to date most SCC studies have focused on assessing susceptibility to SCC initiation in SCW using slow strain rate tensile tests (SSRT) [9].

So, in [13], SSRT tests of unirradiated F/M steel (JLF-1LN) were carried out at 400 and 510 °C at strain rates of 6.4∙10^−4^ s^−1^ and 5∙10^−7^ s^−1^ in supercritical pressurized water of 25 MPa pressure and 8 ppm dissolved oxygen. Scanning electron microscope (SEM) observations revealed that JLF-1LN steels showed a brittle fracture mode in the limited zone close to the specimen surface. This fracture mode is typical for SCC and is caused by the effect of the corrosive environment on the specimen. Further failure of the specimen seems to be due to the loss of carrying capacity.

In [10], the SCC susceptibility of F/M steel (T91) in a supercritical water environment was studied. To investigate the SCC susceptibility of this steel, two types of tests were used: SSRT at strain rates of 1.5∙10^−7^ s^−1^ and 3∙10^−7^ s^−1^, and U-bend at 500, 550, and 600 °C. SEM analysis of the fracture surface of SSRT specimens revealed that T91 steel was not susceptible to SCC in the SSRT tests at 550 and 600 °C. U-bend specimens of T91 did not show any cracking at the apex of the U-bend at either 500 °C or 550 °C.

In [14], the tensile and stress corrosion cracking behavior of ferritic-martensitic steels (T91, HCM12A, and HT-9) in supercritical water were studied. A series of SSRT tests at a strain rate of 3∙10^−7^ s^−1^ were conducted in supercritical water over a temperature range of 400–600 °C and a pressure of 24.8 MPa. Fractography showed that all of the specimens exhibited ductile rupture except for HT-9, which showed evidence of intergranular cracking. Intergranular cracking in HT-9 is affected by temperature and oxygen concentration in supercritical water.

Similar results were obtained in [12]. Ferritic-martensitic alloys have shown to be less susceptible to SCC. The steels HT9, T91, T91a, T92, and T92a HCM12A were tested in pure supercritical water at temperatures ranging from 370 °C to 600 °C with various dissolved oxygen contents. Only HT-9 showed susceptibility to SCC, and its susceptibility was lower than that of the austenitic stainless steels and most of the nickel-base alloys.

Ferritic-martensitic steel HT-9 irradiated with 2 MeV protons at 400 °C and 500 °C to 7 dpa was tested in deaerated SCW [12]. It was shown that the cracking susceptibility of this steel increased with the irradiation level.

So, most of the studies on the SCC of F/M steels were carried out using unirradiated steels. There are few studies of the SCC of austenitic steels irradiated by neutrons [12]. A set of 21 specimens made from Japanese prime candidate alloy (JPCA) (Ti-stabilized 316 SS) were irradiated in the fast flux test facility (FFTF) in the temperature range from 390 °C to 520 °C at doses ranging from 26.9 to 43.9 dpa. These specimens were tested in pure SCW and argon. All specimens tested in SCW were very susceptible to cracking, with significant portions of IG cracking on the fracture surface. Specimens tested in Ar showed no cracks [15,16].

As noted above, stainless F/M steels in the unirradiated state are susceptible to general corrosion but are more resistant to SCC in supercritical environments compared to austenitic steels [17]. This is due to the fact that the thicker iron- and chromium-based oxide layer on the surface of F/M steel is a stronger barrier to the transport of metal ions and oxygen than one for austenitic steels. It is known that chromia-containing spinels are the best barriers for the penetration of cations (metal) and anions (oxygen, OH, etc.) as compared to iron oxides [18]. Chromia-containing spinel is the inner oxide layer that can provide the best corrosion resistance in the SCW environment [17].

At present, there are few data on the effect of irradiation on the susceptibility of stainless F/M steels to SCC in a supercritical environment. In fact, only irradiated HT-9 steel has been investigated in detail in [4]. It was shown that irradiation-induced hardening may be the reason for increased susceptibility to intergranular SCC in HT-9 steel. It has also been shown in this work that grain boundary (GB) carbides may be responsible for the initiation of intergranular cracks. HT-9 steel was susceptible to SCC in both its initial and irradiated states. As mentioned above, most stainless F/M steels are not susceptible to SCC; therefore, the SCC mechanisms in HT-9 steel are apparently not typical for other F/M stainless steels.

Most of the data relates to the SCC of irradiated stainless steel in PWR and BWR environments [19,20,21,22,23,24,25,26,27,28,29]. According to these investigations, the main mechanisms of SSCs are GBs chromium depletion, hardening, localized deformation, GBs weakening, and creep-induced GBs sliding. Perhaps some of these mechanisms are also important for the SCC of irradiated F/M steels in a supercritical environment. Further research is required to determine the role of all these factors in the initiation of SCC in irradiated F/M steels in supercritical environments.

The main mechanisms of degradation during the SWCRs operation of the F/M steels used for the internals are their radiation embrittlement and SCC in the primary circuit coolant environment.

In contrast to austenitic steels, radiation swelling, and high-temperature radiation embrittlement are not the dominant mechanisms of F/M steels degradation under conditions of the internal operation [30].

Thus, SCC can be one of the mechanisms limiting the lifetime of the F/M steels as a material for the SWCR internals. Therefore, it is important to assess the SCC susceptibility of the irradiated F/M steel in SCW in the range of operating temperatures.

So, summarizing the above, this paper studies the SCC susceptibility of neutron-irradiated 12 Cr F/M steel in a supercritical water environment. For this purpose, autoclave tests with a constant load for 500 h are carried out. The results of these tests are used to determine a threshold stress below which the irradiated 12 F/M steel is not susceptible to SCC. Suggestions are made about possible mechanisms of the initiation of SCC in the studied 12Cr F/M steel.

## 2. Material and Experimental Procedure

The chemical composition of 12Cr F/M steel is given in Table 1.

The material under study was irradiated as round bars of 10 mm in diameter and 57 mm in length in the BOR-60 (Dimitrovgrad, Russia) fast-neutron research reactor to various doses approximately equal to 12 dpa at different irradiation temperatures. The irradiation conditions are given in Table 2.

Disk specimens with a diameter of 10 mm and a thickness of 1.5 and 1.9 mm were used for autoclave SCC tests. Specimens were made by cutting the round bars Ø10 mm on the electro-discharge machine “Arta 120”. One side of each specimen was polished. Specimens were loaded in such a way that tensile stresses arise on the polished side. After autoclave tests, this surface was analyzed using a scanning electron microscope (SEM), the VEGA 3 TESCAN, to identify corrosion microcracks.

For a correct comparison of the test results of materials irradiated at different temperatures with different yield strength values, it is necessary to load the disk specimens to provide approximately the same stresses related to the yield strength of the irradiated material.

Disk specimens were placed in the loading device, which provides the necessary load value during the test. The loading was carried out by means of a 6 mm diameter rod on the unpolished side of the disk specimen. The load was supported by a compressed spring, the compliance of which far exceeded the compliance of the specimen. The thicknesses of the specimens were chosen so that, with the same forces acting on the springs, the tensile stresses arising in the specimens corresponded approximately to the same portion of the yield strength of the material irradiated at different temperatures. For this purpose, disk specimens of materials irradiated at different temperatures were made with different thicknesses: 1.5 mm for 390 °C and 1.9 mm for 550 °C.

Then the loading device was placed in the autoclave in an environment simulating the primary circuit coolant of SWCR. The autoclave was designed and manufactured in CRISM “Prometey”. The autoclave scheme is shown in Figure 1. The loading scheme and the loading device are shown in Figure 2 and Figure 3. According to this scheme, loading is carried out in the test machine to a given load. When the disk specimen is loaded with the pusher, the spring is compressed. After loading, the position of the compressed spring is fixed. The spring is made of a special nickel alloy with high creep resistance at autoclave operating temperatures.

During tests, the autoclave water environment was continuously monitored. The parameters in the autoclave, such as pressure, temperature, electric conductivity, pH, and oxygen concentration were monitored. The pH of the environment was maintained in the range of 6.7–6.8. The oxygen concentration throughout the test was less than 1 µg/kg. The electric conductivity of the environment during the tests was in the range of 3–5 μS/cm.

Dependences of pressure and temperature in the autoclave on time are shown in Figure 4.

## 3. Load Calculation for Disc Specimens in the Loading Device

The main task of autoclave tests is to determine the threshold stress below which specimens are not susceptible to SCC.

According to [19,31,32,33], irradiated austenitic stainless chromium-nickel steels are susceptible to SCC in the WWER primary circuit coolant environment at stresses above 0.3–0.4-σ_Y_ (σ_Y_ is the yield strength of irradiated material). Thus, to determine the threshold stress, it is necessary to carry out autoclave tests in the simulated primary circuit coolant environment of SWCR at stresses in the range of 0.3∙σ_Y_÷0.7∙σ_Y_ in order to confirm the corrosive microcrack nucleation at stresses above the threshold and the absence of microcracks at stresses below the threshold.

The loading device installed in the autoclave provides loading of disk specimens according to the scheme shown in Figure 2.

To determine the relationship between the load and the maximum normal tensile stresses, finite element method (FEM) calculations were performed using the ANSYS-14.5 code. The problem was solved in the axisymmetric statement. During modeling, the contact condition between the punch and the specimen was taken into account. To adequately account for changes in the contact area, the problem was solved in a finite-strain statement. Four nodal isoparametric axisymmetric elements were used, the size of which in the contact-modeling zone did not exceed 0.02 mm.

The finite-element approximation for the considered disk specimens is presented in Figure 5.

The punch material and disk material were assumed to be linearly elastic, with Young’s modulus E = 2.0∙10^5^ MPa and Poisson’s ratio ν = 0.3. The calculations showed that varying the friction coefficient in the range from 0 to 0.3 has practically no effect on the stress-strain state of the specimen. Therefore, the friction coefficient was taken to be equal to 0.

Figure 6 shows the dependences of maximum tensile stresses ($\sigma_{\max}=\sigma_{r}^{\max}=\sigma_{\theta }^{\max}$, localized in the center of the disk specimen on the polished surface) on load P. The maximum value of load P was taken to be 1000 N, which corresponds to the maximum force that can be set in the used loading device.

Figure 6 shows that the dependences of σ_max_ on P are linear at *p* < 1000 N. Therefore, for the convenience of calculations, it is proposed to use the dependence in the form:σ_max_ = k_p_∙P,(1)
where the k_p_ values calculated for Figure 6 are presented in Table 3.

The load P acting on the specimens in the autoclave at T = 450 °C was calculated for steel 12Cr F/M using the data from Table 3.

It should be noted that the loading of specimens in the loading device is carried out at T = 20 °C. When heated from 20 to 450 °C, the load from the compressed spring will decrease due to the decrease in Young’s modulus with increasing temperature. Therefore, to provide the required load in the autoclave P_aut_, the given load P_20_ at T = 20 °C is calculated by the formula:(2)$$P_{20}=P_{aut}\cdot \frac{E_{20}}{E_{out}},$$
where E_20_, E_aut_—Young’s modulus values at T = 20 °C and temperature in autoclave accordingly. In this experiment, T_aut_ = 450 °C.

According to [34], $\frac{E_{20}}{E_{aut}}=1.12$ for the nickel alloy from which the spring was made.

Table 4 presents the values of P_20_, P_aut_, at T = 450 °C, σ_max_, and σ_max_/σ_Y_, where σ_Y_ is the yield strength value at T = 450 °C.

The highest susceptibility to SCC for 12Cr F/M steel is expected at its greatest embrittlement, taking place at T_irr_ = 390 °C. Therefore, the largest number of specimens was tested for 12Cr F/M steel irradiated at T_irr_ = 390 °C.

## 4. Autoclave Tests Results

After the autoclave SCC tests, the polished surfaces of the disk specimens were analyzed using a VEGA 3 TESCAN scanning electron microscope (SEM) placed in a “hot” chamber.

The results of SEM examination for specimens № 1, 4, and 7 (T_irr_ =390 °C, Table 4) and № 9–11 (T_irr_ = 550 °C, Table 4) are shown in Figure 7, Figure 8, Figure 9, Figure 10, Figure 11 and Figure 12. All SEM images presented were obtained for the central region of the disk specimen 1000 × 1000 µm (yellow square).

The analysis of Figure 7, Figure 8 and Figure 9 (T_irr_ = 390 °C) shows that the nucleation of microcracks is observed only for disk specimens № 1 and № 4 loaded up to 801 N and 605 N, respectively. For the specimen № 1, the number of such microcracks and the degree of their opening are higher than for the specimen № 4 loaded up to a value of 605 N. On the surface of specimen № 7 microcracks, nucleation is practically not observed. 

Analysis of Figure 10, Figure 11 and Figure 12 (T_irr_ = 550 °C) shows that microcrack nucleation is observed only for disk specimens № 9 and № 10, loaded up to 801 N and 650 N, respectively. On the surface of specimen № 11 microcrack, nucleation is practically not observed.

On the surface of all disk specimens after autoclave tests, an oxide film with a thickness of about 10 µm is observed. At high magnifications, crystals of iron oxides, apparently magnetite and hematite, are observed [35].

For greater crack opening and to create more suitable conditions for quantitative analysis of the nucleated microcracks, all disk specimens were additionally loaded in air by a load of 6023 N (for material irradiated at T_irr_ = 390 °C) and 6100 (for material irradiated at T_irr_ =550 °C) at 20 °C for three minutes and then unloaded. The load values were chosen so as to prevent ductile fracture (Section 5). If all of the observed microcracks nucleated due only to the cracking of the oxide film, then after loading to 6023 N (6100 N), approximately the same number of microcracks with the same opening would be observed on the surface of all disk specimens. Thus, in those cases where the number and opening of microcracks are the same, these microcracks are most likely related to the cracking of the oxide film. Therefore, in further studies, these microcracks are not considered SCC microcracks. If the opening of microcracks depends on the P_aut_ level, then these microcracks are caused by the SCC mechanism.

## 5. Calculation of Specimen Deformation for Microcrack Opening

As it was shown above, the identification of corrosion microcracks after testing specimens in an autoclave is a difficult task because of a thick oxide layer on the specimen surface. One of the simplest methods of corrosion microcrack identification is mechanical loading of specimens, which leads to the opening of microcracks under the oxide layer and, accordingly, to the cracking of the low-strength oxide layer. The loading value should satisfy at least the following conditions: Firstly, the load should not produce tensile plastic deformations that result in microcrack nucleation on brittle or ductile mechanisms. Secondly, the load should not cause significant corrosion or microcrack growth. Thirdly, the load should lead to the corrosion microcracks opening and the oxide layer fracturing.

Based on the above requirements, plastic strain levels at a test temperature equal to 20 °C were determined.

To ensure the absence of ductile fracture, the following level of plastic strain was taken:(3)$${\left(\varepsilon_{r}^{p}\right)}_{max}={\left(\varepsilon_{\Theta }^{p}\right)}_{max}=0.09$$
(4)$${\left(\varepsilon_{eq}^{p}\right)}_{max}=0.18.$$
where ${\left(\varepsilon_{r}^{p}\right)}_{max}$, ${\left(\varepsilon_{\Theta }^{p}\right)}_{max}$, ${\left(\varepsilon_{eq}^{p}\right)}_{max}$ are radial, circumferential, and equivalent plastic strains in the center of the tensile polished surface of the specimen, respectively.

The level of ${\left(\varepsilon_{\mathrm{eq}}^{p}\right)}_{\max}$ was taken based on an estimate of the decrease in fracture strain under biaxial loading typical for disk specimens as compared to fracture strain for a smooth round bar under tension. Then,
(5)$${\left(\varepsilon_{eq}^{p}\right)}_{max}=\frac{\varepsilon_{f}^{biaxial}}{n_{\varepsilon }},$$
where $\varepsilon_{f}^{\mathrm{biaxial}}$ is the fracture-plastic strain in ductile fracture under biaxial loading; n_ε_ is the safety factor introduced to avoid corrosion microcrack growth.

The dependence of ε_*f*_ on the triaxiality of the stress state $\frac{\sigma_{m}}{\sigma_{eq}}$ was estimated by the formula [36], such as the Hancock-Mackenzi equation [37]
(6)$$\varepsilon_{f}^{}=const\cdot \exp\left(-1.5\frac{\sigma_{m}}{\sigma_{eq}}\right),$$

In Equation (6) $\sigma_{m}$ is the hydrostatic stress, $\sigma_{eq}$ is the equivalent stress. Then, assuming that $\frac{\sigma_{m}}{\sigma_{eq}}=\frac{1}{3}$ for a smooth round bar under tension and $\frac{\sigma_{m}}{\sigma_{eq}}=\frac{2}{3}$ for disk specimen tests, it may be calculated:(7)$$\varepsilon_{f}^{biaxial}=\varepsilon_{f}^{uniaxial}\cdot exp\left(-1.5\left(\frac{2}{3}-\frac{1}{3}\right)\right)\approx 0.61\cdot \varepsilon_{f}^{uniaxial},$$
where
(8)$$\varepsilon_{f}^{uniaxial}=-ln\left(1-\frac{\psi }{100}\right),$$
where ψ is the relative area reduction in %.

According to the preliminary studies, the values ψ for 12Cr F/M steel at T_test_ = 20 °C are given in Table 5. In the same table, the values of $\varepsilon_{f}^{\mathrm{uniaxial}}$ and $\varepsilon_{f}^{\mathrm{biaxial}}$ calculated by Formulas (7) and (8) are given.

At ${\left(\varepsilon_{\mathrm{eq}}^{p}\right)}_{\max}$ = 0.18, the safety factor n_ε_ varies from about 2 to 2.5 (see Table 5).

The loading of the irradiated specimens not subjected to the autoclave test confirmed the estimate made by ${\left(\varepsilon_{\mathrm{eq}}^{p}\right)}_{\max}$ = 0.18. In order to confirm the absence of cracking on the surface of the disk specimen not tested in the autoclave; the irradiated specimen was loaded for three minutes in air to a load of 6023 N at room temperature.

The SEM examination of the surface of the specimen not tested in the autoclave is shown in Figure 13.

Figure 13 shows that the surface of the specimen not tested in the autoclave after loading shows signs of plastic deformation and numerous sliding lines. No microcracks are observed. Lines on the surface resembling microcracks (see Figure 13a) are actually traces of plastic deformation (see Figure 13b).

In addition, the subsequent loading of the specimens previously subjected to autoclave tests showed the possibility of microcrack opening and, accordingly, microcrack identification after loading up to ${\left(\varepsilon_{r}^{p}\right)}_{\max}={\left(\varepsilon_{\Theta }^{p}\right)}_{\max}=0.09$ and ${\left(\varepsilon_{\mathrm{eq}}^{p}\right)}_{\max}=0.18$ and subsequent unloading (Section 6).

To estimate the level of load providing the required level of plastic deformation, FEM calculations of the elastoplastic deformation of specimens from 12Cr F/M steel irradiated at 390 °C and 550 °C were performed.

The calculation scheme is presented in Section 3. During the calculation, the stress-strain curve of the material was represented as
(9)$$\sigma_{\mathrm{eq}}=\sigma_{Y}+A{\left(\varepsilon_{\mathrm{eq}}^{p}\right)}^{n},$$
where A and n are strain hardening coefficients.

The values of A and n for T = 20 °C were determined using the tensile test results of smooth round bars. The values σ_Y_, A, and n are presented in Table 6. Figure 14 shows the dependences of $\varepsilon_{\mathrm{eq}}^{p}$ on P at T = 20 °C for 12Cr F/M steel irradiated at T_irr_ = 390 °C and 550 °C.

Load values providing the required plastic deformation for the opening of corrosion microcracks are given in Table 7.

## 6. Additional Loading of Disk Specimens after SCC Tests

Disc specimens № 1–№ 8 (material irradiated at 390 °C) and disk specimens № 9–№ 11 (material irradiated at 550 °C) after autoclave testing were loaded by a load of 6023 N and 6102 N, respectively (as calculated in Section 5), in air at room temperature using a Zwick z050 testing machine.

As an example, the results of SEM examination for specimens № 1, № 4, and № 7 (T_irr_ = 390 °C, Table 4) after SCC tests and additional loading up to 6023 N for three minutes at room temperature are shown in Figure 15, Figure 16 and Figure 17. The results of SEM examination for specimens № 9–№ 11 (T_irr_ = 550 °C, Table 4) after SCC tests and additional loading up to 6102 N for three minutes at room temperature are shown in Figure 18, Figure 19 and Figure 20. All SEM images presented were obtained for the central region of the disk specimen 1000 × 1000 µm (yellow square).

## 7. Discussion

### 7.1. Quantitative Analysis of Microcrack Sizes

As it was shown in Section 6, no microcracking was observed on the surface of the irradiated disk specimen not tested in the autoclave after loading up to 6023 N. Thus, microcracks nucleated on the surface of specimens № 1–№ 8 (material irradiated at 390 °C) and № 9–№ 11 (material irradiated at 550 °C) are related either to the cracking of the oxide film or to the SCC occurrence. As it was mentioned, if the observed microcracks were related exclusively to cracking of the oxide film, the distribution of microcracks in terms of the number and value of their openings would be approximately the same for all specimens and would not depend on the values of loads at autoclave tests. To determine the nature of microcracks arising on the surface of autoclave-tested specimens, their quantitative analysis was performed. Using the ImageJ program (https://imagej.nih.gov/ij/, accessed on 13 March 2023), the value of the microcrack opening, length, and number of cracks were determined on the central section of disk specimens with an area of 1 mm^2^.

SEM examination of the surfaces of disk specimens №1–№8 (material irradiated at 390 °C) and № 9–№ 11 (material irradiated at 550 °C) showed that the greatest number of microcracks has an opening in the range of 4–8 microns. Microcracks of this type were observed in all specimens. Significantly fewer microcracks have an opening in the range of 8–12 μm and 12–20 μm. Such microcracks were observed in the specimens irradiated at 390 °C, which were loaded in the autoclave to loads of 605 N and 801 N. Very few cracks of this type were observed for the specimen irradiated at 550 °C, which was loaded in the autoclave to loads of 356 N.

Thus, for each disk specimen, the total microcrack lengths were determined for the following crack opening ranges: 4–8 μm, 8–12 μm, 12–20 μm. The obtained values are presented as histograms in Figure 21 and Figure 22.

Analysis of Figure 21 and Figure 22 shows that microcracks with an opening of 4–8 µm are observed on the surfaces of absolutely all disk specimens. Apparently, the appearance of these microcracks is most likely associated with the oxide film cracking during additional loading after autoclave testing.

Thus, only microcracks with an opening of more than 8 µm may be related to the SCC occurrence. They are mainly observed on the surface of disk specimens irradiated at 390 °C, № 1–№ 3 (801 N), and in a much smaller amount on the surface of specimens № 4 and № 5 (605 N) (Figure 21). Their number and degree of opening are directly related to the magnitude of the load in the autoclave. Thus, a threshold load at which nucleation of microcracks with an opening smaller than 8 microns takes place is 423 N; i.e., the threshold stress below which cracking does not occur in the environment simulating the primary circuit coolant of SWCR is 296 MPa (0.34σ_Y_), which corresponds to a load of 423 N for 12Cr F/M steel irradiated at 390 °C.

No microcracks with an opening greater than 8 µm are observed for specimens irradiated at 550 °C, with the exception of specimen № 11 (Figure 22). For this specimen, the total length of cracks with openings larger than 8 microns is extremely small, and the presence of such microcracks, apparently, relates to the smaller thickness of specimen № 11 in comparison with specimens № 9 and № 10. As a result, the ${\left(\varepsilon_{eq}^{p}\right)}_{max}$ value for specimen № 11 is larger than for specimens № 9 and № 10, and hence, the microcrack opening for microcracks located in oxide film may be larger too. Thus, we can conclude that the microcracks on disk specimens № 9–№ 11 irradiated at 550 °C are not related to the SCC occurrence but are associated with the oxide film cracking. That is why this 12Cr F/M steel irradiated at 550 °C is not susceptible to SCC at stresses in the range of 135-304 MPa (0.31∙σ_Y_-0.69∙σ_Y_), which corresponds to the range of loads 356-801 N in the environment simulating the primary circuit coolant of SWCR.

### 7.2. Study of the Nature of Microcracks

To confirm that the observed microcracks with an opening larger than 8 μm nucleated and grew into the metal by SCC mechanism, the following additional experimental studies were performed.

In the first step of these studies, the thickness of the oxide layer was determined. In the second step, it was shown that the depth of microcracks with an opening larger than 8 μm (Figure 21 and Figure 22) is much greater than the thickness of the oxide layer. Finally, it has been shown that the zones near the microcrack tip, located in the depth of the microcrack, are the metal, not the oxide layer. These studies are considered in detail below.

To determine the thickness of the oxide layer, the disk specimen after the autoclave SCC test was tested on the testing machine in the brittle regime—at minus 140 °C (Figure 23a). SEM investigation of the fractured surface of broken disk specimens allowed us to reveal two fracture modes. The first mode is microcleavage, typical for the considered F/M steel. The second mode corresponds to the fracture of the oxide layer. The observed difference between the fracture modes allows us to determine the oxide layer thickness. From Figure 23b, it is seen that the thickness of the oxide layer is about 5μm. Analysis of all the images showed that the thickness of the oxide layer varies from 5 to 8 μm.

To confirm that the second mode of fracture corresponds to the oxide layer, an energy-dispersive X-ray spectroscopy analysis (EDS) for points 1 and 2 was performed (Figure 23b).

The AZtec program of “Oxford Instruments Nanotechnology Tools Ltd.” is used to carry out EDS analysis. In all EDS spectra in Figure 23 and Figure 24, the *x*-axis shows the energy of X-ray quanta, and the *y*-axis shows the number of counts per second related to the energy.

The oxygen content at point 1 was 28.5 wt%, while no oxygen was detected at point 2. It means that point 2 lies in the metal below the oxide layer. The distance between points 1 and 2 is about 6 μm. It means that the thickness of the oxide layer is less than 6 μm.

Figure 24 shows the surface of the disk specimen № 1 (T_irr_ = 390 °C, Table 4) after autoclave SCC tests with a load value of 801 N and additional loading up to 6023 N (unbroken specimen). From this figure, it is seen that the depth of microcracks with openings larger than 8 μm exceeds values of 10–12 μm. The performed estimations of microcrack depths give the minimum values because only the projections of depth on the specimen surface may be measured by SEM.

Thus, the depth of microcracks with openings larger than 8 μm exceeds the thickness of the oxide layer. Hence, the microcracks marked by the lines in Figure 24a,b, propagate into the metal, and, therefore, their propagation is related to the SCC mechanism.

An EDS analysis was performed on the surface of the oxide layer and in the depth of the microcrack in Figure 24a. The results are shown in Figure 24c,d.

The oxygen content for point 1 is 29.3 wt% and for point 2 is 1.3 wt%. Point 1 corresponds to the oxide layer and point 2 is located in the depth of the microcrack. There is a sharp decrease in oxygen concentration in the depth of the microcrack. This means that microcracks propagate deeply into the metal by the SCC mechanism (Figure 24a,b).

In Figure 25, the zone of microcrack propagation by the SCC mechanism is shown.

### 7.3. Analysis of Possible SCC Mechanism

The unirradiated 12Cr F/M steel is practically not susceptible to SCC in pressurized water up to temperatures of 550 °C since this steel was created for steam generators of fast-neutron reactors of the BN type. Irradiation leads to a sharp hardening of steel as well as the depletion of grain boundaries by chromium and the development of creep processes in the irradiated material. All these factors induce corrosion-cracking processes.

The experimental results show that the studied F/M steel becomes susceptible to stress corrosion cracking after irradiation at 390 °C but is not susceptible to it after irradiation at 550 °C. Precipitation of chromium carbides, typical for stainless F/M chromium steels, is enhanced with an increase in irradiation temperature [38]. Considering that the SCC susceptibility decreases with the increase in irradiation temperature, chromium carbides are not, at least, the main cause of SCC in the studied F/M steel.

Table 8 shows the values of the Vickers hardness for the studied F/M steel in the initial and irradiated states at different temperatures.

Table 8 shows that the radiation hardening of the steel is practically absent when it is irradiated at 550 °C, while at 390 °C ${\Delta H}_{v}^{0}$ = 1345 MPa. Based on the obtained data on SCC, it can be concluded that radiation hardening strongly affects the susceptibility of steel to SCC. If radiation hardening occurs mainly due to the formation of dislocation loops (which is typical for F/M steels), which are weak barriers for moving dislocations, then such hardening can result in localized deformation [39,40,41].

Radiation-hardened steel in some cases is subjected to the creep process when it is loaded after irradiation [19]. In addition, radiation hardening leads to an increase in the portion of creep strain due to grain boundary (GB) sliding.

Localized deformation, the creep process, and GB sliding, in the general case, can lead to SCC since these processes result in the passivating film rupturing [19].

To clarify the SCC mechanisms, a TEM investigation and creep tests of irradiated 12Cr F/M steel should be provided in subsequent investigations.

## 8. Conclusions

1.Special disk specimens from irradiated 12Cr F/M steel and loading devices for these specimens have been designed for autoclave SCC testing at constant load in an environment simulating the primary circuit coolant of the supercritical water-cooled reactors. The use of such specimens with loading devices allows one to test several specimens with different loading levels in the same autoclave in a supercritical environment.2.Maximum relative stresses in disk specimens tested in the autoclave vary from 0.34σ_Y_ to 0.64∙σ_Y_ for materials irradiated at 390 °C and from 0.31∙σ_Y_ to 0.69∙σ_Y_ for materials irradiated at 550 °C. Such a wide stress range is necessary for the experimental determination of the threshold stress below which the SCC does not occur.
The loads acting on the specimen in the loading device and providing the specified stresses were calculated by FEM.3.Autoclave tests on SCC of disk specimens from 12Cr F/M steel irradiated to the damage dose of 12 dpa at temperatures of 390 °C and 550 °C were carried out at constant loads in the environment simulating the primary circuit coolant of SWCR for a time exceeding 500 h.4.SEM examination of disk specimen surfaces after autoclave tests showed that corrosion microcracks could not be detected due to a sufficiently thick oxide film on the surface of the specimen. Therefore, a method allowing us to improve the detection of corrosion microcracks has been developed. The essence of the method is additional loading of the specimens after autoclave tests. The additional load should increase corrosion microcrack opening as well as oxide film cracking, while at the same time preventing the nucleation of microcracks by ductile or brittle mechanisms.5.After additional loading, microcracks related to the SCC occurrence were detected by SEM on the surface of specimens irradiated at 390 °C and loaded in the autoclave at 605 N and 801 N. Microcracks related to the SCC occurrence were not observed either on the surface of the specimens irradiated at 390 °C and loaded in an autoclave at 423 N, or on the surface of the specimens irradiated at 550 °C. The SCC nature of these microcracks was confirmed by additional mechanical tests, SEM investigations, and EDS analysis of specimen zones located at different depths.6.As a result of the autoclave tests, it was shown:
12Cr F/M steel irradiated at a temperature of 550 °C is not susceptible to SCC at stresses in the range 135÷304 MPa (0.31σ_Y_÷0.69σ_Y_) in the environment simulating the primary circuit coolant of SWCR. When the irradiation temperature is lowered to 390 °C, this steel is not susceptible to SCC at stresses below the threshold stress of 296 MPa (0.34∙σ_Y_). So, for 12Cr F/M steel irradiated in the temperature range (390÷550) °C, the threshold stress may be approximately assumed to be equal to 300 MPa.7.It is shown that the formation of chromium carbides is not the main cause of SCC in irradiated 12Cr F/M steel. The main cause of SCC is the radiation hardening of the steel, which can potentially lead to localized deformation, the creep process, and an increase in the portion of creep strain due to GB sliding.8.To clarify the SCC mechanisms, a TEM investigation and creep tests of irradiated 12Cr F/M steel should be provided in subsequent investigations.

## Figures and Tables

**Figure 1 materials-16-02585-f001:**
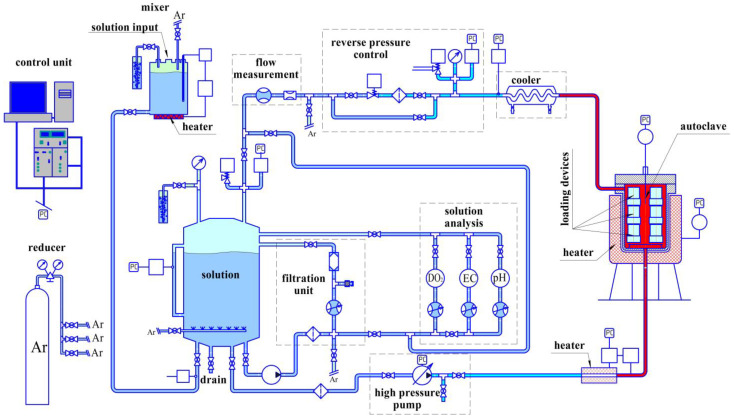
The autoclave scheme.

**Figure 2 materials-16-02585-f002:**
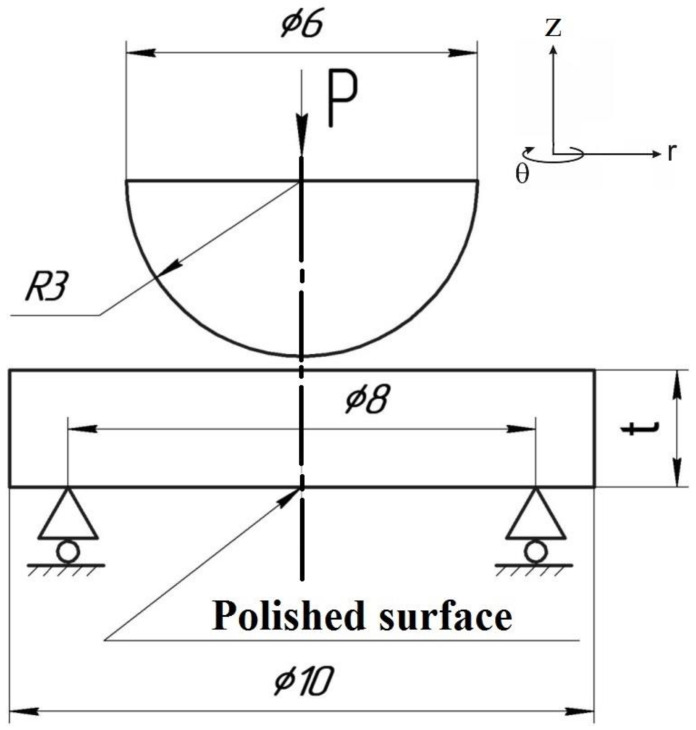
The loading scheme of disk specimens in autoclave. t = 1.5 mm for 12Cr F/M steel, irradiated at T_irr_ = 390 °C; t = 1.9 mm for 12Cr F/M steel, irradiated at T_irr_ = 550 °C.

**Figure 3 materials-16-02585-f003:**
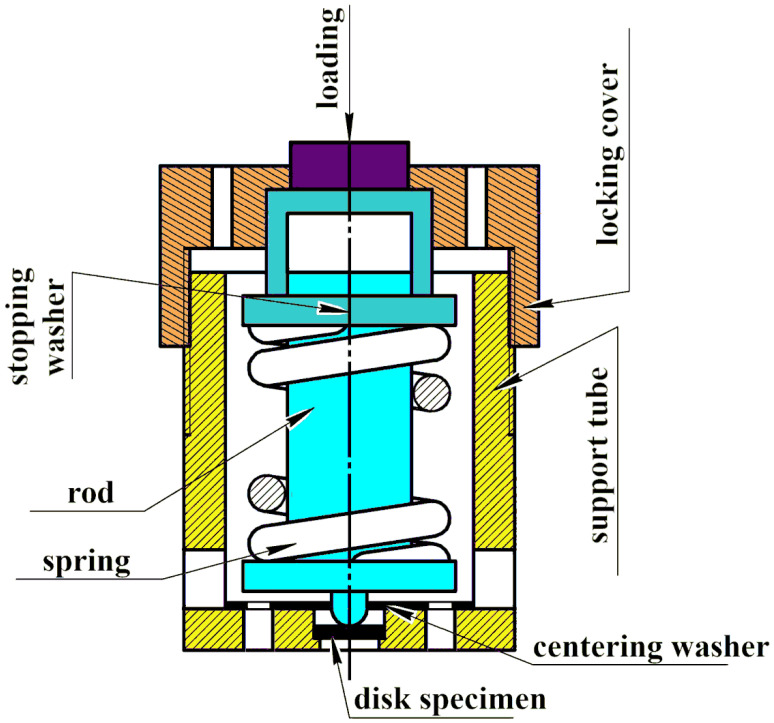
The loading device scheme for autoclave tests of disk specimens.

**Figure 4 materials-16-02585-f004:**
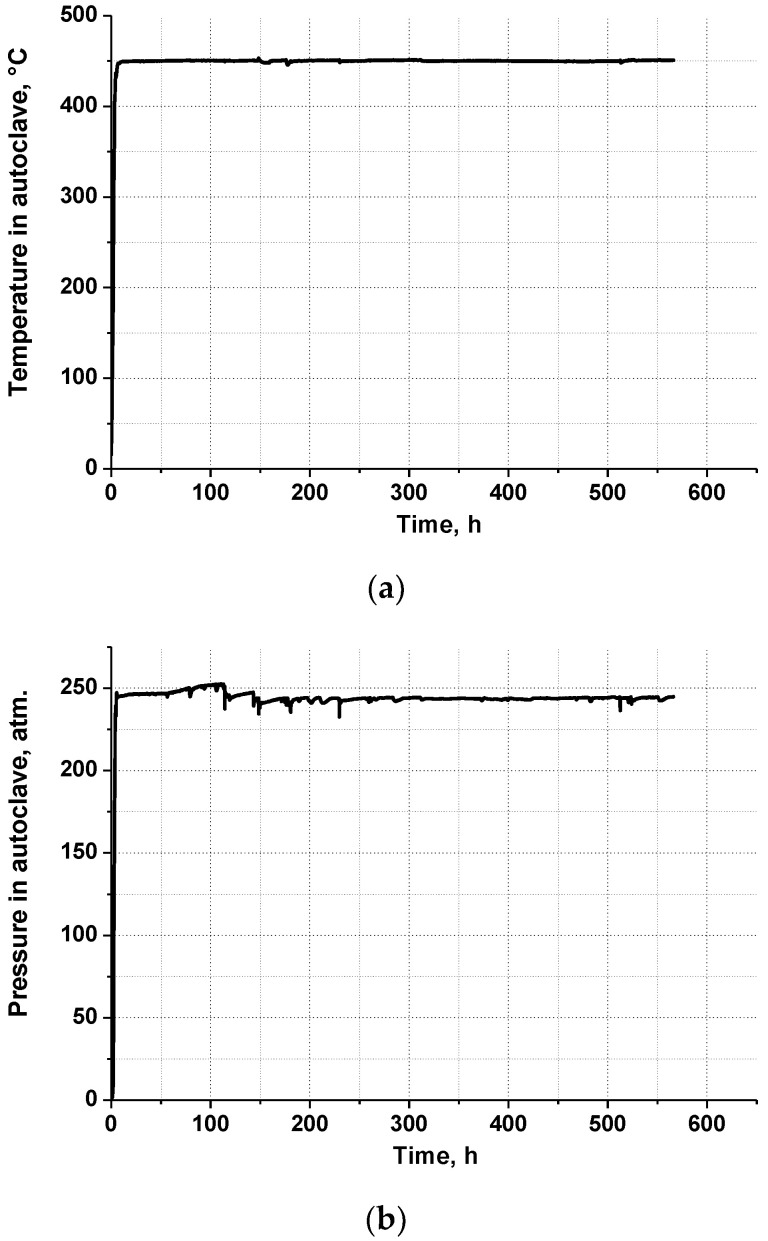
Dependences of temperature (**a**) and pressure (**b**) in the autoclave on time.

**Figure 5 materials-16-02585-f005:**
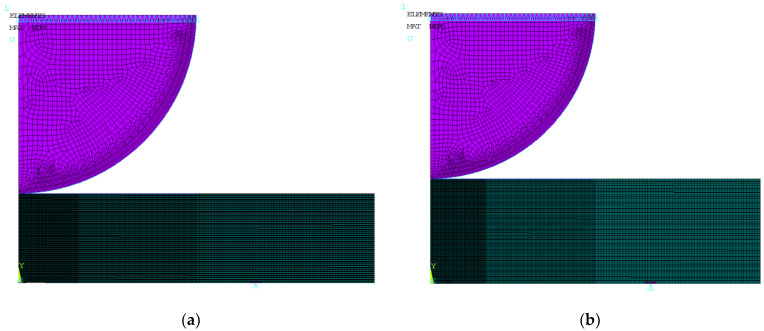
Finite element mesh for a 1.5 mm thick disk specimen made of 12Cr F/M steel irradiated at 390 °C (**a**) and 1.9 mm thick disk specimen made of 12Cr F/M steel irradiated at 550 °C (**b**).

**Figure 6 materials-16-02585-f006:**
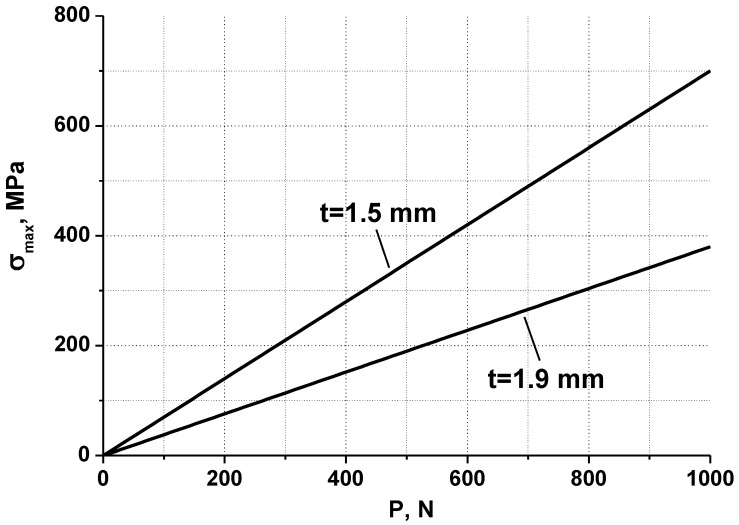
Dependence of $\sigma_{r}^{\max}=\sigma_{\theta }^{\max}$ on P for a specimen of 12Cr F/M steel, at T_irr_ = 390 °C (specimen thickness t = 1.5 mm) and for a specimen of 12Cr F/M steel, at T_irr_ = 550 °C (specimen thickness t = 1.9 mm).

**Figure 7 materials-16-02585-f007:**
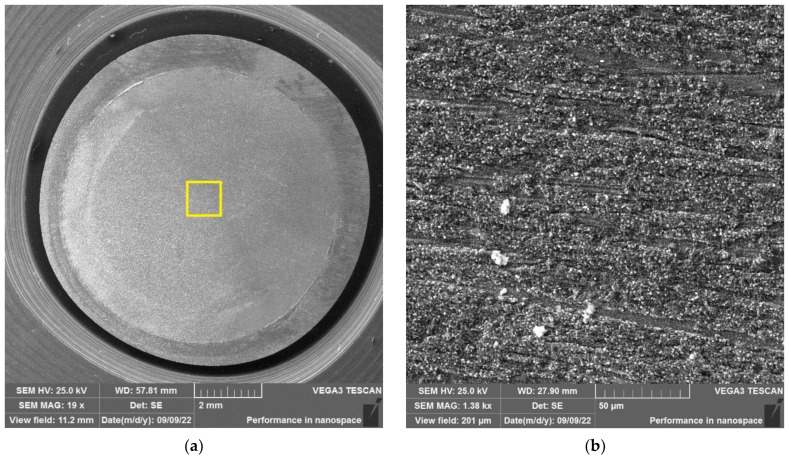
Surface of disk specimen № 1 (T_irr_ = 390 °C, Table 4) at different magnifications after autoclave SCC tests with a load value of 801 N (before crack opening loading).

**Figure 8 materials-16-02585-f008:**
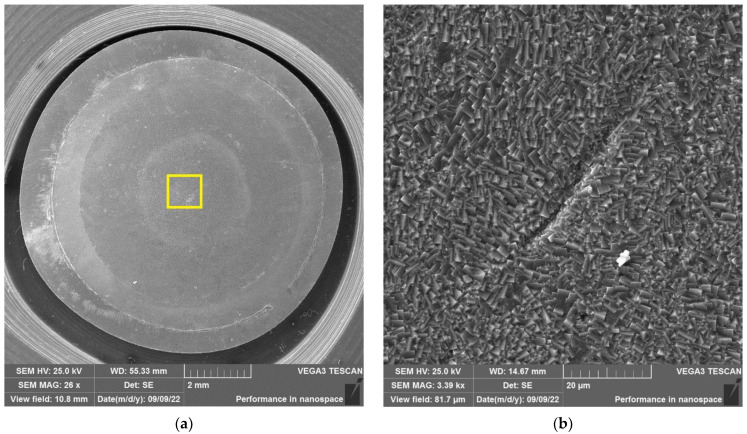
Surface of disk specimen № 4 (T_irr_ = 390 °C, Table 4) at different magnifications after autoclave SCC tests with a load value of 605 N (before crack opening loading).

**Figure 9 materials-16-02585-f009:**
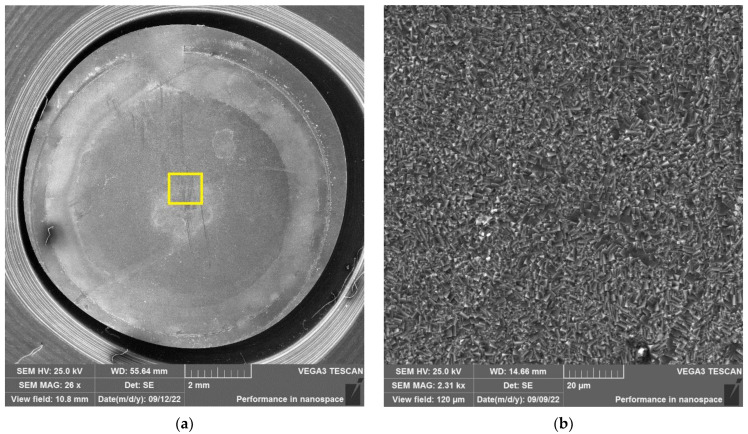
Surface of disk specimen № 7 (T_irr_ = 390 °C, Table 4) at different magnifications after autoclave SCC tests with a load value of 423 N (before crack opening loading).

**Figure 10 materials-16-02585-f010:**
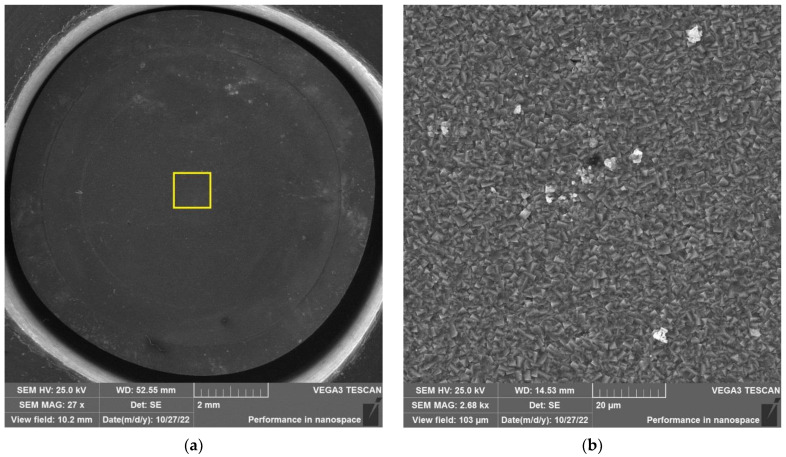
Surface of disk specimen № 9 (T_irr_ = 550 °C, Table 4) at different magnifications after autoclave SCC tests with a load value of 801 N (before crack opening loading).

**Figure 11 materials-16-02585-f011:**
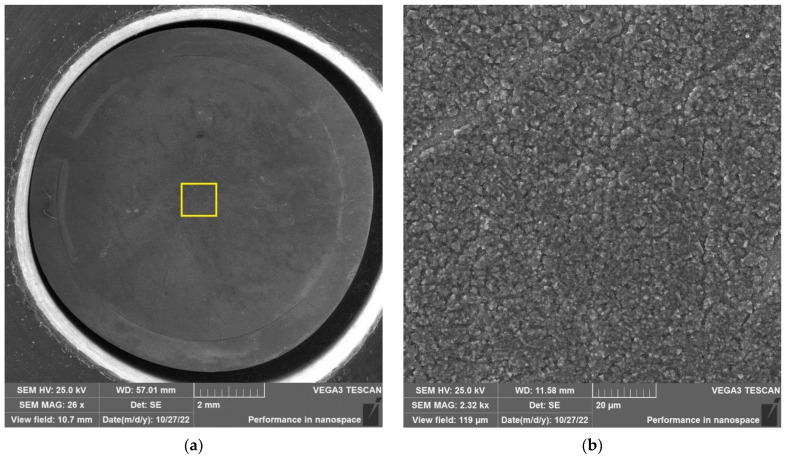
Surface of disk specimen № 10 (T_irr_ = 550 °C, Table 4) at different magnifications after autoclave SCC tests with a load value of 650 N (before crack opening loading).

**Figure 12 materials-16-02585-f012:**
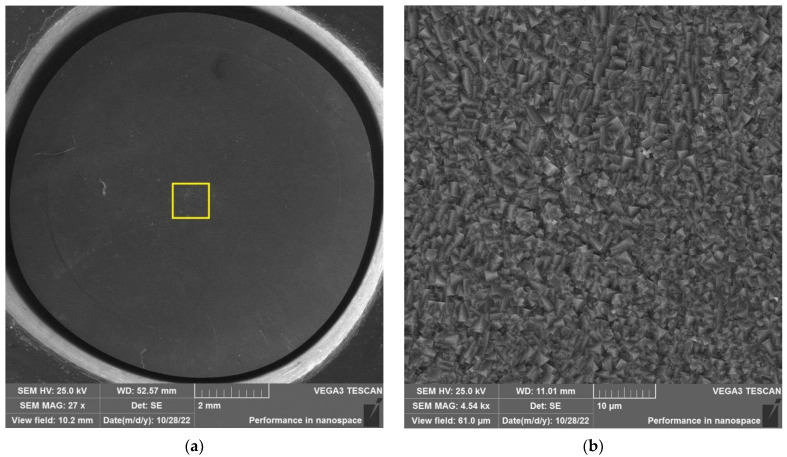
Surface of disk specimen № 11 (T_irr_ = 550 °C, Table 4) at different magnifications after autoclave SCC tests with a load value of 356 N (before crack opening loading).

**Figure 13 materials-16-02585-f013:**
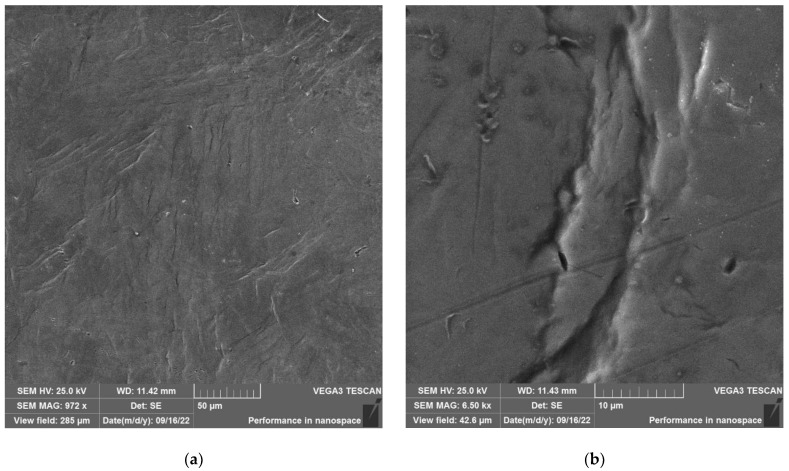
Surface of disk specimen not tested in the autoclave at different magnifications after loading in air at room temperature up to 6023 N.

**Figure 14 materials-16-02585-f014:**
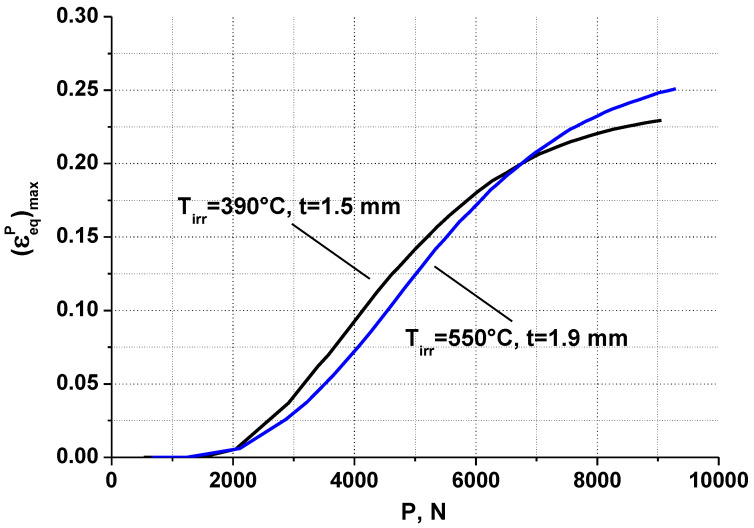
The dependences of ${\left(\varepsilon_{\mathrm{eq}}^{p}\right)}_{\max}$ on P at T = 20 °C for 12Cr F/M steel irradiated at T_irr_ = 390 °C and 550 °C.

**Figure 15 materials-16-02585-f015:**
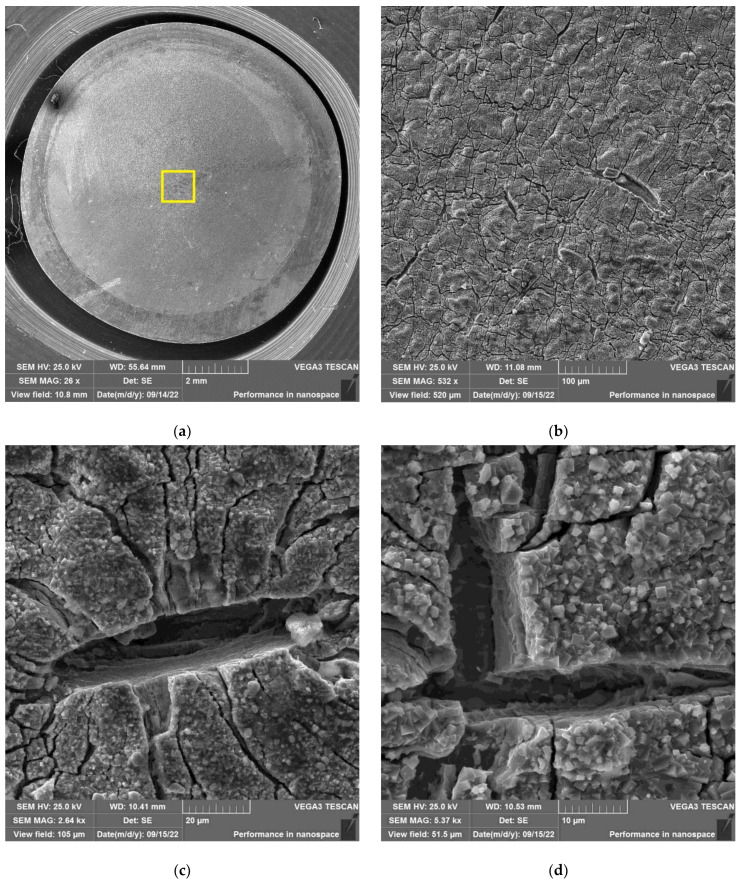
Surface of disk specimen № 1 (T_irr_ = 390 °C, Table 4) at different magnifications after autoclave SCC tests with a load value of 801 N and additional loading up to 6023 N.

**Figure 16 materials-16-02585-f016:**
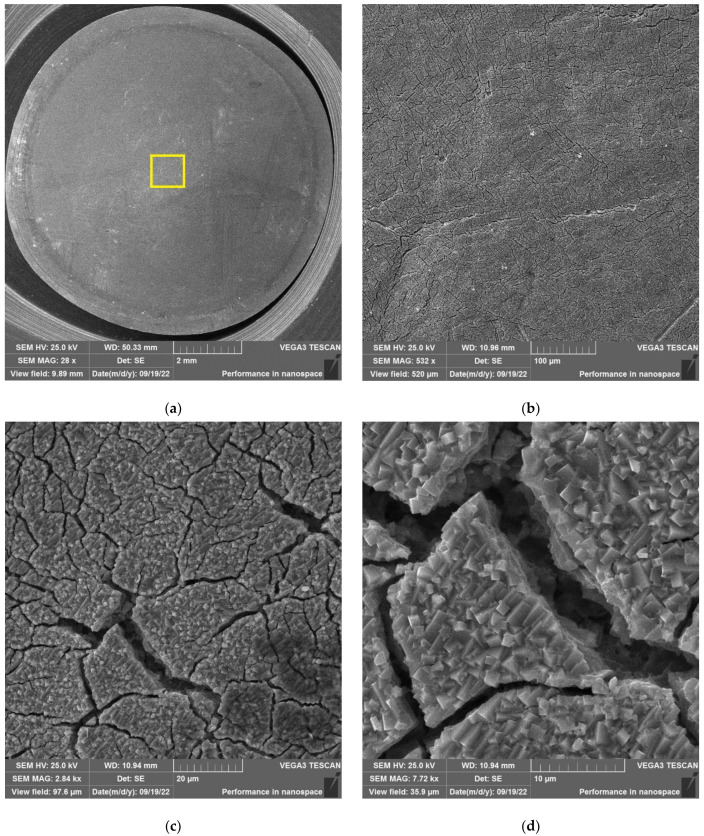
Surface of disk specimen № 4 (T_irr_ = 390 °C, Table 4) at different magnifications after autoclave SCC tests with a load value of 605 N and additional loading up to 6023 N.

**Figure 17 materials-16-02585-f017:**
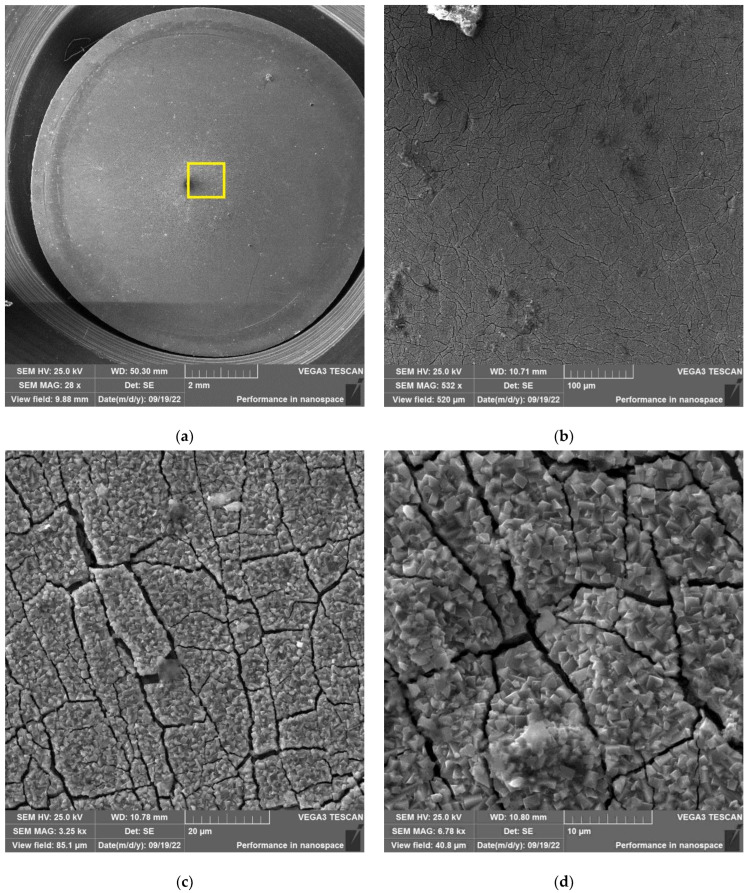
Surface of disk specimen № 7 (T_irr_ = 390 °C, Table 4) at different magnifications after autoclave SCC tests with a load value of 423 N and additional loading up to 6023 N.

**Figure 18 materials-16-02585-f018:**
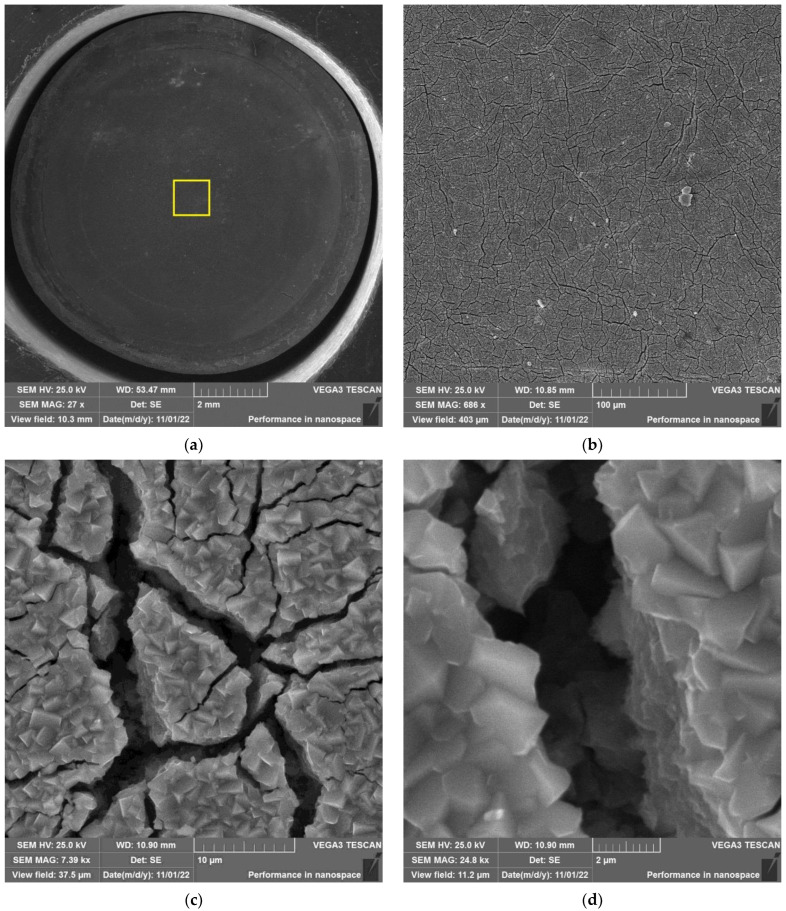
Surface of disk specimen № 9 (T_irr_ = 550 °C, Table 4) at different magnifications after autoclave SCC tests with a load value of 801 N and additional loading up to 6102 N.

**Figure 19 materials-16-02585-f019:**
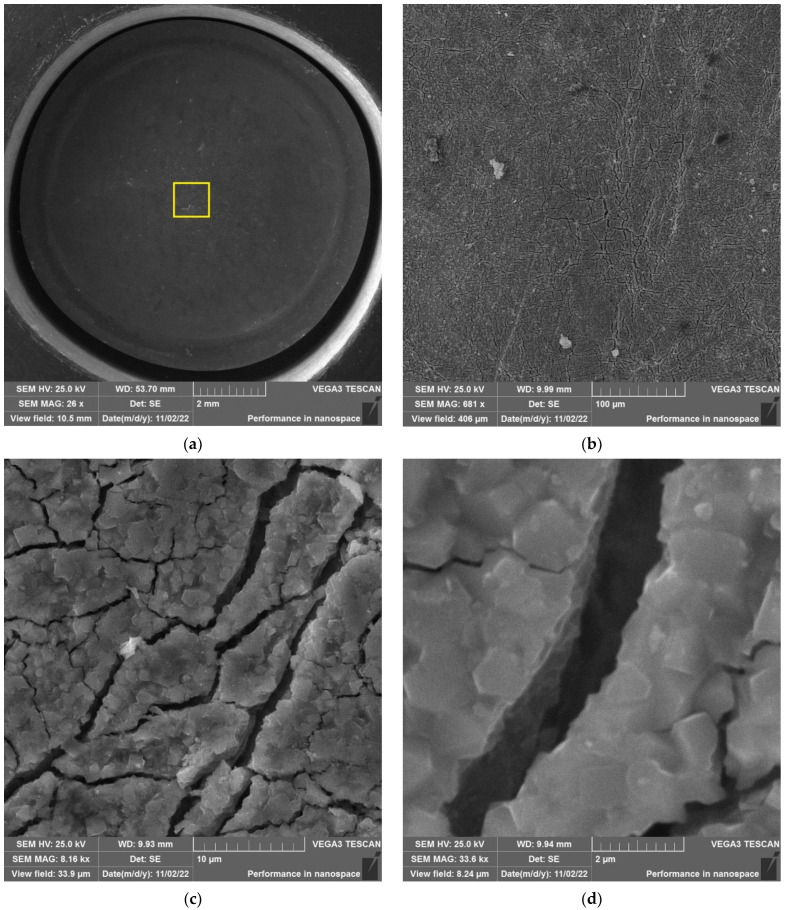
Surface of disk specimen № 10 (T_irr_ = 550 °C, Table 4) at different magnifications after autoclave SCC tests with a load value of 650 N and additional loading up to 6102 N.

**Figure 20 materials-16-02585-f020:**
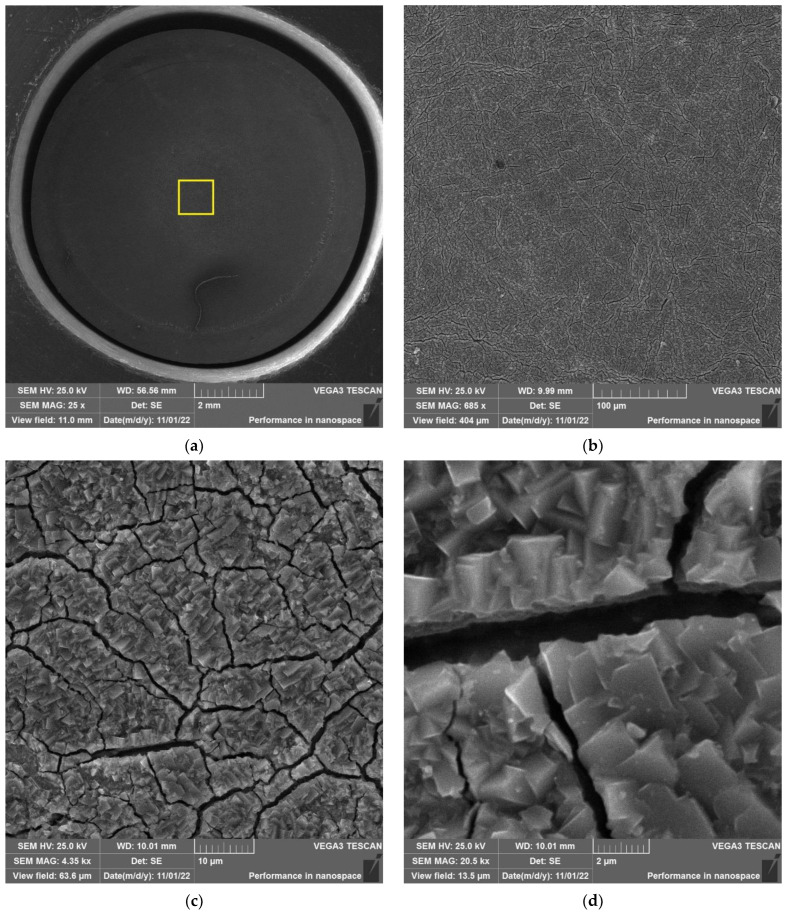
Surface of disk specimen № 11 (T_irr_ = 550 °C, Table 4) at different magnifications after autoclave SCC tests with a load value of 356 N and additional loading up to 6102 N.

**Figure 21 materials-16-02585-f021:**
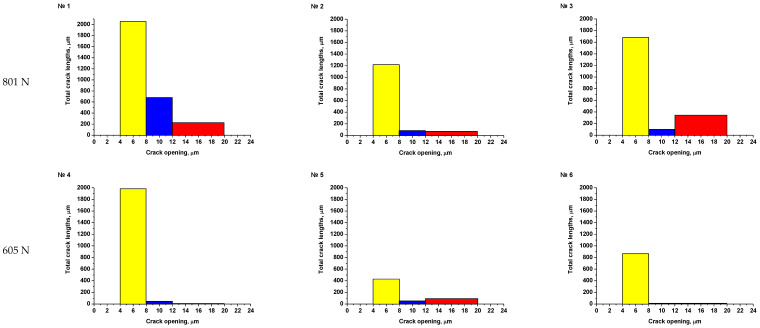
Total microcracks lengths with different microcrack opening values for specimens № 1–№ 8 irradiated at 390 °C after autoclave tests and additional loading.

**Figure 22 materials-16-02585-f022:**
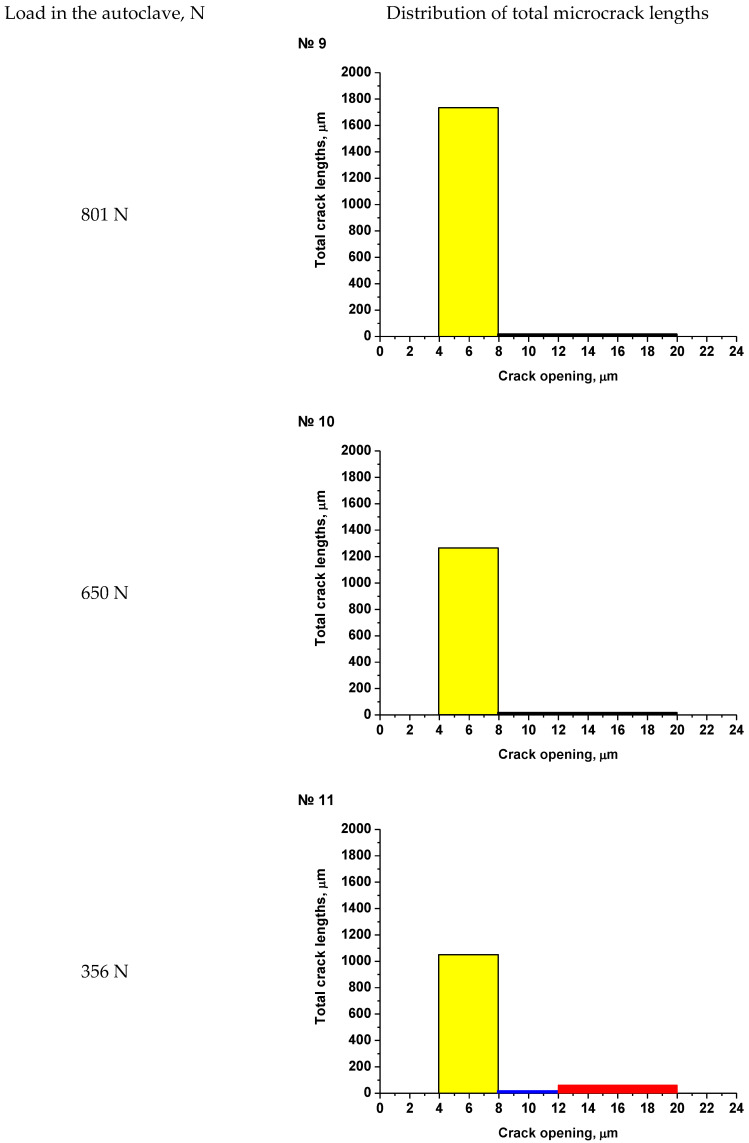
Total microcracks lengths with different microcrack opening values for specimens № 9–№11 irradiated at 550 °C after autoclave tests and additional loading.

**Figure 23 materials-16-02585-f023:**
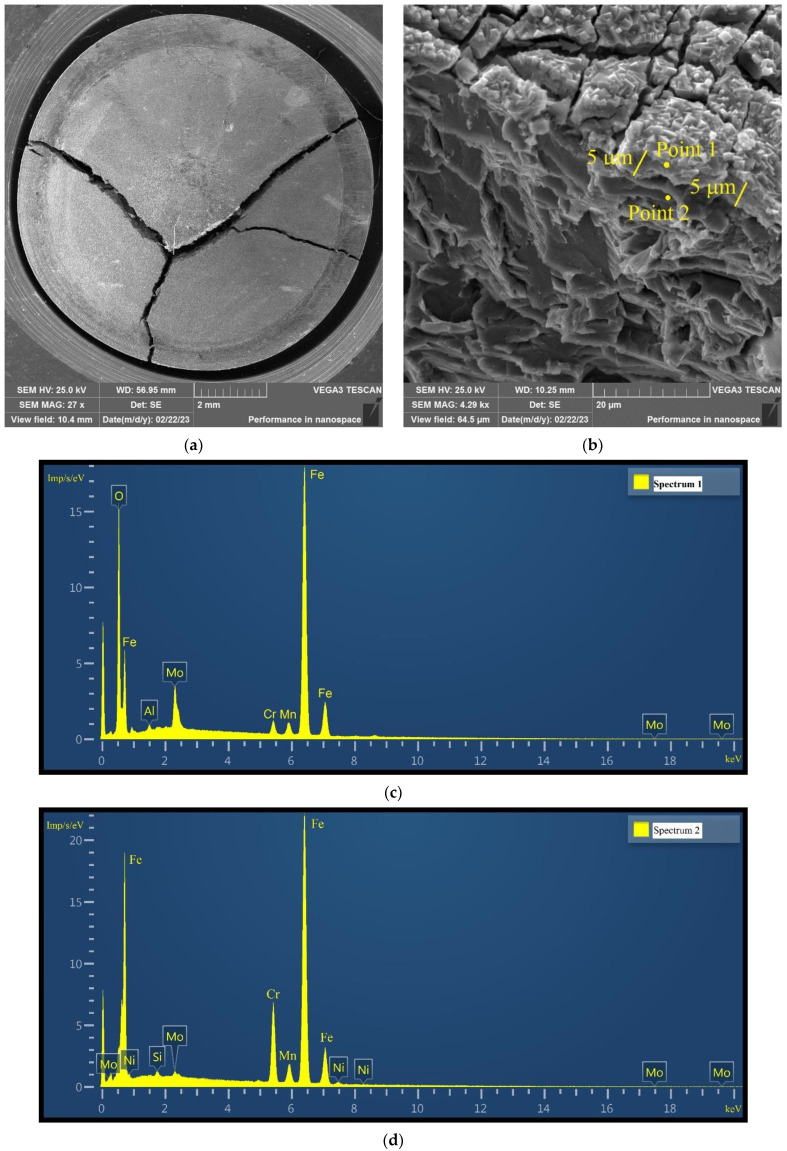
Surface of the disk specimen (T_irr_ = 390 °C) after autoclave SCC test with a load value of 801N fractured at minus 140 °C. An overall view of broken-disk specimen (**a**). Fractured surface of broken-disk specimen(**b**). An oxide layer of ~5 μm thick is observed. Under the oxide layer there is a microcleavage fracture of the metal (12Cr F/M steel) (**b**). EDS spectra for points 1 (**c**) and 2 (**d**).

**Figure 24 materials-16-02585-f024:**
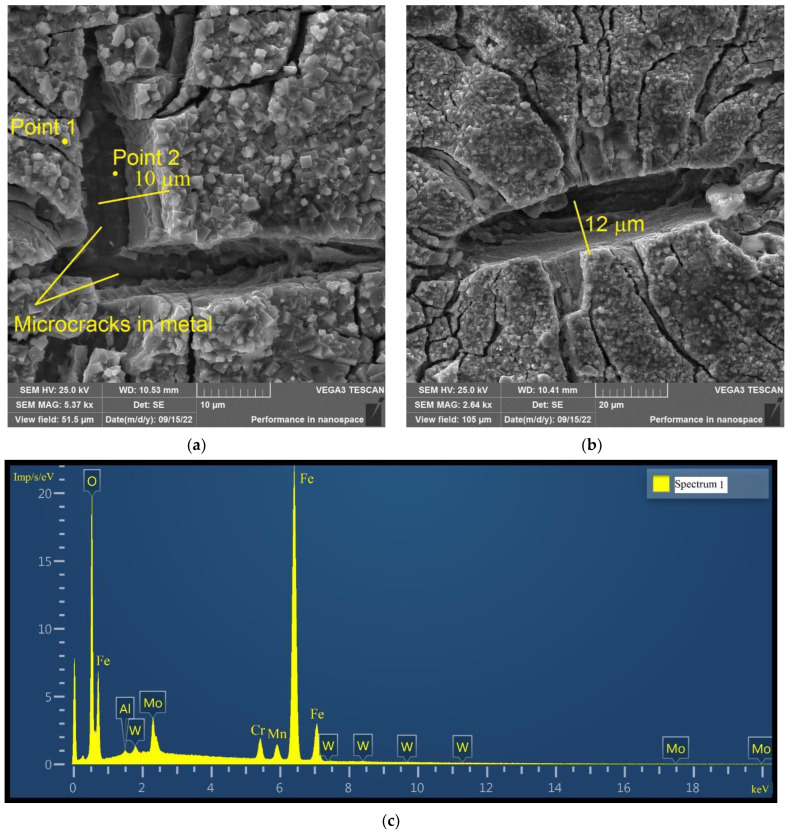
Surface of disk specimen № 1 (T_irr_ = 390 °C, Table 4) at different magnifications after autoclave tests with a load value of 801 N and additional loading up to 6023 N (unbroken specimen). Microcracks propagating into metal by SCC mechanism are marked by lines (**a**), (**b**); microcracks depth exceeds 10–12 μm. EDS spectra for points 1 (**c**) and 2 (**d**).

**Figure 25 materials-16-02585-f025:**
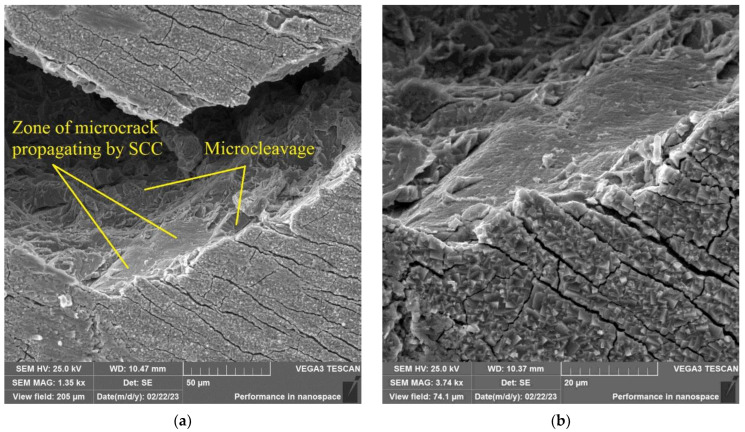
Surface of disk specimen (Tirr = 390 °C) after autoclave SCC tests with a load value of 801 N and subsequent fracture at minus 140 °C. Zone of microcrack propagating by SCC mechanism at different magnifications (**a**,**b**) is shown. Microcleavage mode typical for brittle fracture of F/M steel is shown too.

**Table 1 materials-16-02585-t001:** The chemical composition of 12Cr F/M steel, wt. %.

C	Si	Mn	Cr	Ni	Mo	Nb	V	N	Al	B	S	P
0.08	0.29	0.70	12.10	1.06	0.94	0.11	0.20	0.059	0.029	0.003	0.003	0.013

**Table 2 materials-16-02585-t002:** Irradiation condition.

Material	Damage Dose, dpa	Irradiation Temperature, °C	Round Bar Dimensions, mm
12Cr F/M steel	11.4	390	Ø10 × 57
	12.9	550	Ø10 × 57

**Table 3 materials-16-02585-t003:** The values of the parameter k_p_ in Equation (1).

Material	t, mm	k_p_, MPa/N
12Cr F/M T_irr_ = 390 °C	1.5	0.70
12Cr F/M T_irr_ = 550 °C	1.9	0.38

**Table 4 materials-16-02585-t004:** The values P_20,_ P_aut_, σ_max_, σ_y_, σ_max_/σ_y_ and t values for 12Cr F/M steel.

Material	t, mm	№	P_20_, N	P_aut_, N	σ_max_, MPa	σ_Y_ (T = 450 °C), MPa	σ_max_/σ_Y_
12Cr F/M steel T_irr_ = 390 °C	1.5	1	900	801	561	870	0.64
		2	900	801	561		0.64
		3	900	801	561		0.64
		4	680	605	424		0.48
		5	680	605	424		0.48
		6	680	605	424		0.48
		7	475	423	296		0.34
		8	475	423	296		0.34
12Cr F/M steel T_irr_ = 550 °C	1.9	9	900	801	304	440	0.69
		10	730	650	247		0.56
		11	400	356	135		0.31

**Table 5 materials-16-02585-t005:** The values ψ, $\varepsilon_{f}^{\mathrm{uniaxial}}$, and $\varepsilon_{f}^{\mathrm{biaxial}}$ for 12Cr F/M steel irradiated at different temperatures.

Material	T_test_, °C	ψ	$\varepsilon_{f}^{uniaxial}$	$\varepsilon_{f}^{biaxial}$	n_ε_
12Cr F/M steel T_irr_ = 390 °C	20	0.45	0.59	0.36	2.00
12Cr F/M steel T_irr_ = 550 °C	20	0.52	0.73	0.45	2.50

**Table 6 materials-16-02585-t006:** The values t, σ_Y_, A, n, E, and ν used for FEM calculations at T = 20 °C.

Material	t, mm	σ_Y_, MPa	A, MPa	n	E, MPa	ν
12Cr F/M, T_irr_ = 390 °C	1.5	1073	244	0.36	2.0∙10^5^	0.3
12Cr F/M T_irr_ = 550 °C	1.9	597	472	0.33	2.0∙10^5^	0.3

**Table 7 materials-16-02585-t007:** Load values providing the required plastic deformation for the opening of corrosion microcracks.

Material	t, mm	T_test_, °C	σ_Y_, MPa	P, Providing ${\left(\varepsilon_{eq}^{p}\right)}_{max}=0.18,$ N
12Cr F/M steel T_irr_ = 390 °C	1.5	20	1073	6023
12Cr F/M steel T_irr_ = 550 °C	1.9	20	597	6102

**Table 8 materials-16-02585-t008:** Conditions of neutron irradiation ($T_{\mathrm{irr}}^{n}$, D), hardness values in the initial, $H_{v}^{0}$, and irradiated, $H_{v}^{\mathrm{irr}}$, states and increment of hardness due to irradiation, ${\Delta H}_{v}^{n}$, of 12Cr F/M steel.

Temperature and Damage Dose	$$H_{v}^{\mathrm{irr}},\;\mathrm{MPa}$$	$$H_{v}^{0},\;\mathrm{MPa}$$	$${\Delta H}_{v}^{n},\;\mathrm{MPa}$$
$T_{\mathrm{irr}}^{n}$ = 390 °C, D = 10.3 dpa	3650	2370	1280
$T_{\mathrm{irr}}^{n}$ = 390 °C, D = 11.6 dpa	3780	2370	1410
$T_{\mathrm{irr}}^{n}$ = 550 °C, D = 14.6 dpa	2380	2370	10

## Data Availability

Not applicable.

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
