# Peer review of "Investigation of Stress Corrosion Cracking Resistance of Irradiated 12Cr Ferritic-Martensitic Stainless Steel in Supercritical Water Environment"

_materials, 2023, doi:10.3390/ma16072585_

Round 1
Reviewer 1 Report
Studying IASCC of structure material in SCW is very challenging. A critical issue that must be addressed is: were the cracks real SCC cracks? FM steel can form thick oxide layer on the sample surface. So the author should show that if the crack extends to the substrate and how the crack propagate. These evidence is so critical to the quality of this paper that I can't approve its publication until they are provided.
Author Response
Thank you very much for your helpful comment. We performed an additional experiment to clarify the thickness of the oxide layer and to confirm that observed microcracks propagate by SCC mechanism.
More detailed explanations have been added to the section 7.2 on p. 30 of the new version of the article.
Reviewer 2 Report
The manuscript under consideration entitled „Investigation of stress corrosion cracking resistance of irradiated 12Cr ferritic–martensitic stainless steel in supercritical water environment” by Margolin et al. raises important and current issues, it is properly prepared in accordance with scientific standards. Therefore I recommend the article for publication in Materials journal with the following comments:
1. In the introduction, it should be more strongly articulated what is a scientific novelty in this manuscript.
2. When giving a pressure value, the unit 'atm.' is not used in currently accepted standards, the unit 'bar' or 'Pa' should be used.
3. The FEM numerical model is insufficiently described, there is no detailed information on how exactly the contact between the materials was defined. The properties of the elements used in the FEM model should be described in more detail. It should be justified why in the FEM modeling friction coefficient equal to 0 was adopted.
4. FEM analysis results are only reliable when validated. Here, there is no information whether the results of the numerical analysis were validated in any way.
5. A more scientific discussion of the obtained research results should be carried out, which should include an in-depth analysis of the reasons for the occurrence of specific phenomena that are presented. The authors only state that a given phenomenon has been observed, for example, on SEM images, such as microcracks, but a scientific discussion is required as to what was the cause of the observed phenomena.
6. SEM images at various magnifications are shown. Higher magnification areas should be marked on SEM macro images because their location must be known.
7. Conclusions are too general. It is necessary to compare results obtained with results from references.
8. Conclusions should contain plans for further research.
9. The formatting does not fully comply with the guidelines imposed by the journal.
10. Old references should be replaced by more recent journal paper references.
11. Editing of English language and style required.
Author Response
Thank you very much for your helpful comment. We have made changes to the article in accordance with your comments.
- Information about scientific novelty have been added to the Introduction on p. 3 (the last paragraph of the Introduction) of the new version of the article.
- The unit for pressure value have been changed by MPa.
- More information on modeling the contact between the materials has been added on pages 7, 8 of the new version of the article.
- The FEM analysis was validated with regard to the bending of disk specimens. The FEM analysis results were compared with the analytical solution for a disk specimen supported along the contour and loaded with pressure on disk surface. The error obtained did not exceed 2.6 %.
- Analysis of the reasons of SCC in supercritical water environment for un-irradiated and irradiated stainless steels is added in the Introduction on p. 3 and to the Discussion (Section 7.3) on p. 34 of the new version of the article.
- All SEM images presented were obtained for the central region of the disk specimen with sizes 1000x1000 µm, where the load was maximum. This information has been added on p. 10, 20 of the new version of the article.
- In the unirradiated state, the studied 12Cr F/M steel, as well as most of F/M steels, is not sensitive to SCC. It is difficult to compare the results obtained with the results from references for irradiated F/M steels because there is data only for proton-irradiated HT-9 steel. But HT-9 steel was susceptible to SCC as in initial as in irradiated state. Thus, the SCC mechanisms of HT-9 steel apparently are not typical for other F/M stainless steels. This information was added in Introduction on p.3 of the new version of the article.
- Plans for further research were added to Discussion on p. 35 and to Conclusions on p. 36 of the new version of the article.
- The formatting has been changed in the new version of the article.
- New more recent journal paper references were added to Introduction and Discussion.
- We tried to improve our English as much as we could.
Reviewer 3 Report
I have gone through the manuscript titled “Investigation of stress corrosion cracking resistance of irradiated 12Cr ferritic-martensitic stainless steel in supercritical water environment”. Overall the work is good and the layout of the manuscript is well-prepared. However, there are comments if worked upon can make the article better and more interesting for the readers.
# Comment 1: All figure captions require more elaboration. They need to be explicit and self-explainable.
# Comment 2: The discussion section must be rewritten. What is the role of irradiation on SCC? A comparison with non-irradiated 12Cr F/M steels will be nice in the discussion.
#Comment 3: Recent articles in this field need to be cited both in the introduction and discussion.
There are a few typo mistakes. It will be nice to correct those also.
Author Response
Thank you very much for your helpful comment. We have made changes to the article in accordance with your comments.
- Figure captions of Fig.3 (p.6), Fig.21 (p.28), Fig.22 (p.29) were improved in the new version of the article.
- An analysis of the role of irradiation on SCC mechanism of the studied steel and comparison with non-irradiated 12Cr F/M steel were added to the Introduction (p.2-3) and Discussion (Section 7.3, p.34-35) of the new version of the article.
- New more recent journal articles references were added to Introduction and Discussion of the new version of the article.
The typos we noticed have been corrected in the new version of the article.
Round 2
Reviewer 1 Report
it can be accepted now.
Reviewer 3 Report
Authors have done the required changes in the revised manuscript.